# Roles of Natriuretic Peptides and the Significance of Neprilysin in Cardiovascular Diseases

**DOI:** 10.3390/biology11071017

**Published:** 2022-07-06

**Authors:** Hitoshi Nakagawa, Yoshihiko Saito

**Affiliations:** 1Cardiovascular Medicine, Nara Medical University, Kashihara 634-8522, Nara, Japan; hitoshi.nakagawa@naramed-u.ac.jp; 2Nara Prefecture Seiwa Medical Center, Mimuro 636-0802, Nara, Japan

**Keywords:** natriuretic peptide, neprilysin, cardiac remodeling, heart failure

## Abstract

**Simple Summary:**

The endocrine effects of atrial natriuretic peptide (ANP) and brain natriuretic peptide (BNP) in the vasculature, and the autocrine effects of ANP and BNP in cardiomyocytes are mediated by the common guanylyl cyclase A receptor (GC-A) expressed in various tissues and cell types. C-type natriuretic peptide (CNP) has paracrine actions that regulate vascular resistance and moderate myocardial stiffness via guanylyl cyclase B receptor (GC-B). Genetically modified mice have revealed the physiological roles of ANP and BNP in blood pressure, cardiac remodeling, and acute myocardial infarction. Molecular pathways in GC-A signaling specifically in cardiomyocytes were also investigated. ANP and BNP via the GC-A signaling phosphorylate regulator of G-protein signaling subtype 4 (RGS4) result in the inhibition of Gαq signaling coupled with angiotensin II type 1A receptor, inhibit the activation of transient receptor potential C6 (TRPC6), and attenuate genomic actions of the cardiac mineralocorticoid receptor (MR). Moreover, recent studies showed the physiological roles of CNP via GC-B in blood pressure and cardiac stiffness. Since natriuretic peptides are degraded by neprilysin (NEP), inhibiting NEP activity is expected to enhance the actions of natriuretic peptides. Experimental studies and clinical trials have shown the effect of NEP inhibition on cardiac remodeling, acute myocardial infarction, and hypertension.

**Abstract:**

Atrial natriuretic peptide (ANP) and brain natriuretic peptide (BNP) activate the guanylyl cyclase A receptor (GC-A), which synthesizes the second messenger cGMP in a wide variety of tissues and cells. C-type natriuretic peptide (CNP) activates the cGMP-producing guanylyl cyclase B receptor (GC-B) in chondrocytes, endothelial cells, and possibly smooth muscle cells, cardiomyocytes, and cardiac fibroblasts. The development of genetically modified mice has helped elucidate the physiological roles of natriuretic peptides via GC-A or GC-B. These include the hormonal effects of ANP/BNP in the vasculature, autocrine effects of ANP/BNP in cardiomyocytes, and paracrine effects of CNP in the vasculature and cardiomyocytes. Neprilysin (NEP) is a transmembrane neutral endopeptidase that degrades the three natriuretic peptides. Recently, mice overexpressing NEP, specifically in cardiomyocytes, revealed that local cardiac NEP plays a vital role in regulating natriuretic peptides in the heart tissue. Since NEP inhibition is a clinically accepted approach for heart failure treatment, the physiological roles of natriuretic peptides have regained attention. This article focuses on the physiological roles of natriuretic peptides elucidated in mice with GC-A or GC-B deletion, the significance of NEP in natriuretic peptide metabolism, and the long-term effects of angiotensin receptor-neprilysin inhibitor (ARNI) on cardiovascular diseases.

## 1. Introduction

The presence of granules in atrial myocytes observed by electron microscope similar to those in endocrine cells was reported in 1956 [1]. Subsequently, the presence of a natriuretic substance in atrial myocytes was reported in 1981 by de Bold [2]. Then, its amino acid sequence was independently reported by two groups in December 1983 [3] and January 1984 [4], and this polypeptide was called the atrial natriuretic peptide (ANP). Brain natriuretic peptide (BNP) was discovered in the porcine brain in 1988 [5] and C-type natriuretic peptide (CNP) in 1990 [6]. ANP and BNP are secreted from the heart, the former mainly from the atria and the latter from the ventricles [7,8,9]. Since the secretion of ANP and BNP from the failing heart is increased [10,11], plasma levels of ANP and BNP are clinically used as a biomarker to diagnose heart failure. ANP induces natriuresis via direct renal effects and vasodilatation, inhibits adrenal aldosterone production, and enhances endothelial permeability [12,13]. ANP and BNP activate the common guanylyl cyclase A receptor (GC-A) expressed in various tissues and cell types. GC-A is a transmembrane receptor containing an intracellular guanylyl cyclase domain. It synthesizes the second messenger cyclic guanosine monophosphate (cGMP) upon ANP/BNP binding to its extracellular domain, which counter-regulates various pressor actions of the renin-angiotensin-aldosterone and the sympathetic nervous system [13,14]. CNP is critical during bone development, which stimulates physiological endochondral bone growth via the autocrine effects of its specific cGMP-producing guanylyl cyclase B receptor (GC-B) in chondrocytes. In the cardiovascular system, CNP is produced mainly in the endothelium and possibly in smooth muscle cells, cardiomyocytes, and cardiac fibroblasts. CNP has paracrine actions that regulate vascular resistance and moderate myocardial stiffness via GC-B [13,15]. Genetically modified mice have revealed the physiological actions of natriuretic peptides via GC-A or GC-B in various cell types. These include the endocrine effects of ANP/BNP in the vasculature, autocrine effects of ANP/BNP in cardiomyocytes, and paracrine effects of CNP in the vasculature and cardiomyocytes.

The metabolism of natriuretic peptides is affected by degradation by the natriuretic peptide clearance receptor (NPR-C), excretion from the kidney, and degradation by neprilysin (NEP). NEP is a neutral endopeptidase distributed throughout the body, including the kidneys, lungs, brain, endothelium, and heart, and degrades natriuretic peptides [16,17,18]. Therefore, inhibiting NEP activity is expected to enhance the actions of natriuretic peptides [19,20,21]. Clinical trials have shown that angiotensin receptor-neprilysin inhibitor (ARNI) improved the prognosis of patients with heart failure [22] and reduced blood pressure in patients with uncontrolled hypertension [23]. Since the clinical importance of the action of natriuretic peptides related to ARNI has been reported, the physiological roles of natriuretic peptides have regained attention. This article focuses on the physiological roles of natriuretic peptides revealed in mice carrying GC-A or GC-B deletion, the significance of NEP in the metabolism of natriuretic peptides, and the long-term effects of ARNI.

## 2. Physiological Roles of Natriuretic Peptides: Lessons from Genetically Modified Mice

### 2.1. Effect of Natriuretic Peptides on Blood Pressure

ANP knockout (KO) mice show salt-sensitive hypertension [24], whereas global GC-A KO mice exhibit salt-insensitive hypertension (an increase of 27 mmHg) with ventricular hypertrophy and fibrosis [25]. An explanation for the acquisition of salt sensitivity in ANP KO mice but not in GC-A KO mice remains unclear.

Intriguingly, smooth muscle cell (SMC)-specific GC-A KO mice have normal arterial blood pressure. Intravenous ANP administration caused a rapid decrease in blood pressure in floxed GC-A mice but not in SMC GC-A KO mice [26]. This indicates that GC-A is not necessary for SMC to regulate blood pressure in chronic conditions, whereas it is critical in the acute blood pressure moderation by ANP.

In contrast, endothelial cell (EC)-specific GC-A KO mice exhibit significantly increased systolic blood pressure (by approximately 12–15 mmHg), which is salt-resistant, compared with those of the control mice. Moreover, the total plasma volume was increased by 11–13% in EC-specific GC-A KO mice [27]. In comparison, the plasma volume of global GC-A KO mice was chronically increased by approximately 30% [28]. ANP infusion caused immediate increases in hematocrit and microvascular albumin permeability in the control but not in EC-specific GC-A KO mice. ANP-induced albumin permeability was mediated by transendothelial caveolae [29]. Endothelial GC-A may regulate the blood pressure by maintaining chronic intravascular volume homeostasis via albumin transport, which could partially explain the mechanism of salt-resistant hypertension in EC-specific GC-A KO and global GC-A KO mice.

Regarding CNP/GC-B signaling, vascular SMC-specific GC-B KO mice showed normal blood pressure similar to those of the control mice. CNP-induced acute vasorelaxation was nearly completely abolished in the mesenteric arteries of SMC-specific GC-B KO mice. In contrast, EC-specific CNP KO mice displayed significantly increased blood pressure values (by approximately 10 mmHg). The gene expression of endothelin-1 (ET-1) was enhanced in pulmonary vascular endothelial cells from EC-specific CNP KO mice. In EC-specific CNP KO mice, the plasma ET-1 concentrations were significantly higher and blood pressure was significantly reduced in response to an endothelin receptor antagonist than their control littermates. This suggests that, for the regulation of blood pressure in chronic conditions, CNP plays an important role in inhibiting ET-1 production through its actions on vascular endothelial cells [30]. Recently, Kuhn et al. developed genetically modified mice with GC-B receptor deletion in the endothelium or pericytes. GC-B deletion in endothelial cells did not change the dilating effects of CNP on the microvasculature. In contrast, deleting GC-B in pericytes abolished this effect of CNP and increased peripheral resistance and arterial blood pressure in chronic conditions. The average increase in blood pressure in pericyte GC-B KO mice was similar to that in EC-specific CNP KO mice. The authors concluded that endothelial paracrine CNP activates GC-B/cGMP signaling in microcirculatory pericytes, thereby lowering the peripheral vascular resistance and arterial blood pressure [31].

### 2.2. Effect of Natriuretic Peptides on Cardiac Remodeling

The ANP concentration in the human heart tissue is extremely high [32]. Therefore, the GC-A receptor in the heart tissue is thought to be activated by ANP and BNP as an autocrine effect to prevent adverse cardiac remodeling. Since global GC-A KO mice show cardiac hypertrophy and interstitial fibrosis [25], the importance of GC-A signaling in cardiac remodeling is recognized. Intriguingly, double KO of global GC-A and global angiotensin II type 1A receptor (AT1R) significantly improved cardiac hypertrophy and interstitial fibrosis. This indicates that GC-A signaling attenuates adverse cardiac remodeling by inhibiting AT1R [33]. Furthermore, Tokudome et al. reported that locally secreted natriuretic peptide-induced activation of the regulator of G-protein signaling subtype 4 (RGS4) inhibits Gαq signaling coupled with AT1R [34]. ANP/PKG signaling also prevents cardiac remodeling by phosphorylating transient receptor potential C6 which activates the calcineurin-NFAT pathway [35]. 

The effect of endogenous ANP and BNP on cardiac GC-A receptor was more evident in cardiomyocyte (CM)-specific GC-A KO mice, which demonstrated an increased heart weight and cardiomyocyte hypertrophy compared with wild-type (WT) mice [36]. Transverse aortic constriction (TAC) significantly induced left ventricular hypertrophy, interstitial fibrosis, and impaired cardiac function in CM-specific GC-A KO mice compared to WT mice, supporting that ANP and BNP produced from the heart directly affect the heart as an autocrine effect, thereby preventing cardiac remodeling. As a mechanism of autocrine or paracrine effects of natriuretic peptides, we observed that GC-A/cGMP/PKG signaling counter-regulates cardiac mineralocorticoid receptor (MR) activation locally in cardiomyocytes [37] (Figure 1). Moreover, aldosterone-induced cardiac remodeling was exacerbated in high-salt conditions in GC-A KO mice independent of blood pressure, whereas these changes were not observed in WT mice. This indicates that GC-A signaling prevents adverse cardiac remodeling by inhibiting the deleterious salt action on aldosterone-induced MR genomic effects in the heart [38]. 

Recently, a CM-specific GC-B KO mouse model was developed. The CNP-induced activation of GC-B/cGMP/PKGI signaling in cardiomyocytes prevented titin-based myocyte stiffening by phosphorylating titin springs at the specific Ser_4080_ site during the early phases of pressure overload in mice. The authors also observed that fibroblasts, endothelial cells, and pericytes might be local sources of CNP three days after TAC. However, TAC-induced hypertrophy and fibrosis did not differ between CM-specific GC-B KO and WT mice [39]. It is important to elucidate whether GC-B in other cell types in the heart, such as fibroblasts, vascular cells, and immune cells, improves adverse cardiac remodeling.

### 2.3. Effect of Natriuretic Peptides on Acute Myocardial Infarction (AMI)

The biological significance of natriuretic peptides following AMI has been studied in terms of their physiological and pathological actions. When GC-A KO and WT mice were applied to coronary ligation permanently, GCA KO mice exhibited more severe ventricular fibrosis in non-infarcted area and post-infarct hypertrophy than in WT mice four weeks after AMI [40]. Moreover, GC-A KO mice, which were subjected to AMI, died within seven days after ligation, probably because of heart failure accompanied by significantly increased lung weight. In contrast, the acute pro-inflammatory roles of ANP and BNP after AMI have been investigated. Two days after MI, a 30-min coronary artery ligation followed by reperfusion, the infarcted area was significantly smaller in GC-A KO than in WT mice. A histological examination revealed low neutrophilic infiltration into the infarct region, with lower P-selectin expression levels in the coronary artery endothelium of GC-A KO mice [41]. BNP-Tg mice, in which the liver-specific human serum amyloid P component promoter was used and the plasma BNP concentration was approximately 3 nmol/L, more frequently died approximately 3–5 days after the ligation of the left coronary artery because of ventricular-free wall rupture, with more massive infiltration of neutrophils in the infarcted area than in WT mice. Immunostaining revealed that a higher expression of metalloproteinase-9 (MMP-9) was observed in infiltrated neutrophils in BNP-Tg mice [42].

A recent study showed that myocardial neutrophil activation and infiltration during the early phase post-AMI were significantly reduced in EC-specific GC-A KO mice. The AMI-dependent induction of endothelial adhesion molecules, such as E/P selectins, was significantly attenuated in EC-specific GC-A KO mice. Furthermore, the authors demonstrated that the expression of phosphodiesterase 2A (PDE2A) in human umbilical vein endothelial cells (HUVEC) was enhanced by tumor necrosis factor-α (TNF-α) and hypoxia, and they observed a significant induction of PDE2A expression in the heart after AMI. Using the cremaster model, in which TNF-α was pretreated to induce PDE2 expression, both ANP and BNP (100 nmol/L) had proinflammatory effects with leukocyte extravasation. The authors concluded that a high expression of PDE2 can mediate the hyperpermeability, which is induced by the effects of endothelial GC-A/cGMP signaling, and it may facilitate neutrophil extravasation during the sub-acute phase after AMI [43]. In some clinical studies, plasma levels of natriuretic peptides in sub-acute phase may be associated with infarct size or cardiac rupture [44,45].

Collectively, pathological plasma levels of natriuretic peptides may induce cardiac rupture at an early phase due to neutrophil invasion in the heart tissue, probably because they induce vascular hyperpermeability in the impaired endothelium. In contrast, natriuretic peptides prevent adverse cardiac remodeling in the chronic phase after AMI.

## 3. Neprilysin Inhibitor and Revival of Natriuretic Peptides

### 3.1. Role of Neprilysin and Significance of Its Inhibition

NEP, also known as acute lymphoblastic leukemia antigen CD10 [46], was discovered in the proximal tubule brush border of rabbits in 1973 [47]. It is a neutral endopeptidase and is classified as a metalloprotease in which zinc is involved in the catalytic mechanism. NEP is a transmembrane protein with an active site in the extracellular domain that cleaves substrates on the amino side of hydrophobic residues [48]. NEP degrades peptides of up to approximately 40 amino acid residues; it has more than 50 types of peptide substrates, including angiotensin II, endothelins, adrenomedullin, bradykinin, and natriuretic peptides [16,49]. NEP is distributed throughout the body, including the kidneys, lungs, brain, endothelium, and heart [16,17,18]. Systemic NEP KO mice exhibit hypotension, low cardiac weight, and increased vascular permeability [50]. However, there have only been a few reports on the use of genetically modified mice in the cardiovascular field. Therefore, the precise roles of neprilysin in each tissue and its effects on many substrate peptides remain unclear.

In clinical trials, treatment with NEP inhibitors showed limited effects on hypertension and heart failure, although the cardiovascular protective effects of elevated natriuretic peptides and bradykinin were expected. Elevation of angiotensin II and ET-1 levels in the plasma may partially contribute to the negative results. Omapatrilat, a molecule that inhibits angiotensin-converting enzyme (ACE), and NEP, was developed. Omapatrilat did not reduce mortality and hospitalization for heart failure compared with ACE inhibitors. Notably, the secondary endpoint (i.e., cardiovascular admission or death from any cause) was significantly improved by 9% [51]. However, angioedema was observed, possibly due to bradykinin production by ACE and NEP inhibition. Thus, omapatrilat was not approved by the Food and Drug Administration in the USA.

LCZ696, which combines the angiotensin receptor blocker (ARB) valsartan with the NEP inhibitor sacubitril, was developed. As valsartan suppresses the action of angiotensin II enhanced by the NEP inhibitor, it can be expected that increased natriuretic peptides have protective effects on the cardiovascular system. ARNI is the first oral drug to enhance the action of natriuretic peptides through this mechanism.

### 3.2. Natriuretic Peptides as a Substrate of Neprilysin

NEP presents a preferential degradation of human natriuretic peptides following the order CNP > ANP > BNP [52,53]. The half-life of ANP and CNP in human plasma is 2–4 min [54,55] and that of BNP is 12 min [54] or ≥20 min [56]. Pancow et al. incubated murine or human natriuretic peptides (10 µmol/L) with recombinant mouse NEP at 37 °C and analyzed the supernatants using high-performance liquid chromatography. The degradation rate of these peptides depended on the N- and C-termini length. Therefore, CNP and human and murine ANP were the best substrates. Notably, human BNP, which has a longer N-terminus than ANP and CNP, was resistant to NEP. Importantly, recombinant human NEP showed the same pattern as the mouse enzyme [53].

The primary NEP cleavage sites in ANP and CNP are between cysteine and phenylalanine, breaking the ring and inactivating the peptide. In contrast, in BNP, it occurs between methionine and valine [52]. Natriuretic peptides enter the internal cavity of NEP. Due to the short amino- and carboxy-terminal tails of ANP and the lack of carboxy-terminal tails of CNP, ANP and CNP can be fixed optimally in NEP and interact with the catalytic cleavage site for the ring at the cysteine–phenylalanine bond. On the other hand, the long tails of BNP do not allow its ring to make a proper substrate position for catalysis, resulting in initial cleavage outside the ring structure at residues methionine-valine [17,53]. Therefore, BNP is a poor substrate for human NEP. These different biochemical effects of neprilysin on three natriuretic peptides may be one of the factors affecting the different half-lives of them.

In a clinical trial, the plasma levels of ANP were measured after ARNI was administered to patients with heart failure with reduced ejection fraction (HFrEF). The mean ANP baseline level of 99 pg/mL increased to 203 pg/mL at three months [57]. Since a decrease in the N-terminal (NT)-prohormone BNP (NT-proBNP) demonstrated improvement in heart failure at three months, it was considered that ANP concentration was further increased immediately after administration. In contrast, in the PARADIGM-HF clinical trial, the BNP levels were mildly elevated after administering ARNI to patients with HFrEF, despite a decrease in NT-proBNP. Since the degree of BNP increase was smaller than that of ANP, the significance of BNP as a NEP substrate was possibly less meaningful than that of ANP, as demonstrated in some in vitro studies. The present BNP immunoassays measure not only active form of BNP but also proBNP [58]. Therefore, the evaluation of active form of BNP after ARNI administration is needed to be elucidated.

### 3.3. Effect of Neprilysin on Cardiac Remodeling

A large clinical trial involving patients with heart failure (NYHA II–IV, ejection fraction ≤40%) using LCZ696 reported that the LCZ696 group reduced the composite outcome of cardiovascular death and heart failure hospitalization by 20% compared to that in the enalapril group [22]. Increased natriuretic peptides by NEP inhibition were thought to act as a hormone on the target organs, resulting in an excellent effect of ARNI. Furthermore, in recent years, the effect of ARNI on reverse cardiac remodeling has been reported.

In the PROVE-HF study, in which ARNI was administered to patients with HFrEF, improvement in left ventricular contractility was observed at 12 months, demonstrating the effect of reverse remodeling by the ARNI [59]. Moreover, a retrospective study was conducted in patients with cancer therapy-related cardiac dysfunction (CTRCD), who were excluded in PARADIGM-HF when cancer chemotherapy was applied within a year. The relatively low dose of ARNI was switched from ACE-inhibitor or ARB, and it significantly improved left ventricular ejection fraction at 4.6 months [60]. However, as mentioned earlier, since the local ANP concentration even in the ventricle is approximately 100–1000 times higher than that in the plasma, a mild increase of the natriuretic peptide in the plasma may not be sufficient to elicit an effect on the heart tissue. Moreover, we previously reported that the local action of ANP against the heart tissue is vital to prevent cardiac remodeling via cardiomyocyte ANP/GC-A/PKG signaling. Thus, the action of cardiac NEP against local ANP in the heart tissue may be important for cardiac remodeling. We generated cardiomyocyte-specific NEP transgenic mice, in which NEP was overexpressed in cardiomyocytes, and investigated the effects of NEP on the cardiac tissue after TAC. TAC significantly exacerbated left ventricular hypertrophy, induced cardiac interstitial fibrosis, and impaired contractile function in cardiomyocyte NEP transgenic mice compared with wild type mice. Since NEP activity in the heart was extremely enhanced in cardiomyocyte NEP transgenic mice, the plasma levels of internal mouse ANP were significantly reduced. Therefore, we administered human ANP exogenously to reach plasma levels of ANP physiologically similar to those in wild-type mice. However, TAC-induced cardiac hypertrophy, interstitial fibrosis, and dysfunction were not rescued by human ANP administration in cardiomyocyte NEP transgenic mice [61]. This suggests that cardiac NEP locally regulates natriuretic peptides in the heart tissue. Notably, the plasma concentration of ANP is in the pM range and increases to a maximum of 1 nmol/L in patients with heart failure. In contrast, in failing heart tissue, the concentration of ANP is around 10–100 nmol/L in the left ventricle. As the Michaelis-Menten constant (km) of NEP for ANP and BNP is approximately 10–100 μmol/L [62,63], NEP may degrade ANP and BNP in heart tissue more efficiently than in plasma. Therefore, ARNI may directly affect the heart and lead to reverse remodeling by increasing cGMP signaling, even at high concentrations of cardiac ANP and BNP [Figure 2]. These protective actions are possibly affected by inhibiting MR, AT1R, and TRPC in heart tissue. Further studies are needed whether augmentation of natriuretic peptides signaling is enough to suppress MR and AT1R activity similar to MR antagonist and ARB in the heart.

### 3.4. Effect of Neprilysin Inhibitor on AMI

Since global GC-A KO mice with permanent occlusion of coronary artery showed worse cardiac remodeling and prognosis than WT mice in the chronic phase [40], it was hypothesized that ANP infusion may improve the prognosis in patients with AMI. Kitakaze et al. conducted the J-WIND study, prospective, randomized, single-blind, a multicenter study to evaluate the efficacy of intravenous ANP compared with nicorandil, in the acute phase in patients with AMI who underwent percutaneous coronary intervention. In the ANP group, the infarct size significantly decreased by 15%, and the ejection fraction was significantly higher [64]. 

In an experimental study, LCZ696 was administered in rats 7 days after left coronary artery occlusion. LCZ696 prevented cardiac remodeling and the impaired left ventricular contraction compared with vehicle at 4 weeks [65]. However, the protective effects of LCZ696 beyond the ACE inhibitor or ARB in vivo study were needed to be elucidated. A recent study has shown that LCZ696 (20 mg/kg body weight) administered to WT mice with permanent occlusion of the left anterior descending artery one day after AMI significantly prevented cardiac rupture compared with mice treated with enalapril (4 mg/kg body weight) by decreasing the MMP-9 mRNA expression and activity in the infarcted area [66]. The authors suggested that the simultaneous regulation of two neurohumoral systems (inhibition of the renin-angiotensin system and enhancement of the natriuretic peptides system) may suppress MMP-9 activity and have protective effects on cardiac rupture after AMI.

A randomized clinical trial (PARADISE-MI) was conducted to evaluate the long-term effects of LCZ696 on AMI. Patients with reduced ejection fraction (LVEF ≤ 40%) or pulmonary congestion were randomly assigned to receive the ARNI or the ACE inhibitor ramipril within 0.5 to 7 days after AMI. Patients were followed-up for an average of 22 months. The composite outcome of cardiovascular death, first hospitalization for heart failure, and first outpatient episodes of symptomatic heart failure was not significantly different between the ARNI and ACE inhibitor groups [67]. However, in an exploratory analysis, the composite outcome of cardiovascular death and total hospitalization for heart failure had a significantly lower relative risk (0.79 in the ARNI group) [68]. 

### 3.5. Effect of Neprilysin Inhibitor on Hypertension

The hormonal effects of ANP and BNP and the paracrine effects of CNP via coordination between endothelial cells and pericytes are important for regulating blood pressure. As mentioned previously, ANP and BNP induce natriuresis, regulate circulating plasma volume via endothelial permeability effect, suppress aldosterone production, and inhibit sympathetic nervous activity. These hormonal effects enhanced by ARNI may contribute to reduction of blood pressure. Natriuresis by ARNI was confirmed in patients with salt-sensitive hypertension. ARNI significantly increased natriuresis at 6 and 24 h on day 1 after first dose administration compared with valsartan but not on day 28, leading to significant reduction of blood pressure on day 28 [69].

The antihypertensive effects of LCZ696 on hypertension reported in clinical studies are consistent. In a study involving patients with mild or moderate essential hypertension, LCZ696 significantly reduced blood pressure after eight weeks compared with valsartan [70]. In a phase III study of patients with essential hypertension uncontrolled by olmesartan, LCZ696 significantly reduced blood pressure at eight weeks compared with olmesartan [23]. Another phase III study was conducted to assess the efficacy and safety of ARNI in Japanese patients with mild to moderate essential hypertension. ARNI significantly reduced systolic blood pressure at 8 weeks compared with Olmesartan [71]. In Japan, sacubitril/valsartan was approved in 2021 to treat patients with hypertension. It would be interesting to determine whether ARNI can prevent the progression of heart failure in patients with asymptomatic hypertensive heart disease.

## 4. Conclusions

Since the structures of natriuretic peptides were determined, natriuretic peptides have been clinically used for treatment of acute heart failure and for biomarkers. The precise roles of natriuretic peptides in each tissue were revealed by genetically modified mice. Recent clinical application of ARNI has brought up the long-term beneficial effects of natriuretic peptides in patients with heart failure. However, the effects of ARNI on angiotensin II, endothelin, amyloid beta, and other substrates are needed to be elucidated in the future.

## Figures and Tables

**Figure 1 biology-11-01017-f001:**
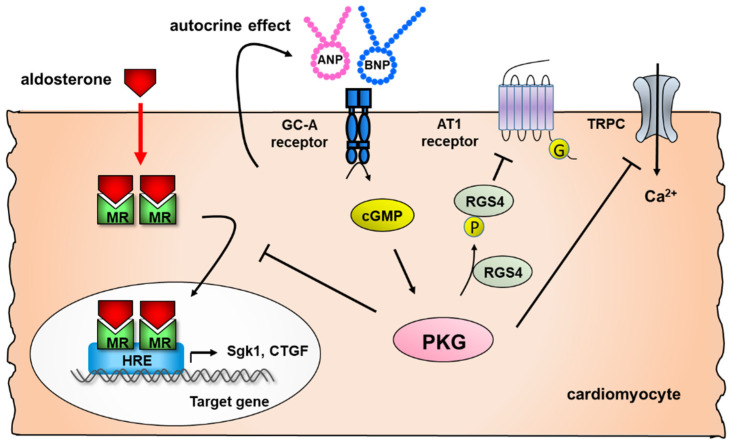
Protective autocrine effects of natriuretic peptides against adverse cardiac remodeling in cardiomyocytes. Natriuretic peptides/GC-A signaling phosphorylates regulator of G-protein signaling subtype 4 (RGS4), resulting in the inhibition of Gαq signaling coupled with Ang II type 1A receptor [33,34], inhibits the activation of transient receptor potential C6 (TRPC6) [35] and attenuates the genomic actions of the cardiac mineralocorticoid receptor (MR) in cardiomyocytes [37].

**Figure 2 biology-11-01017-f002:**
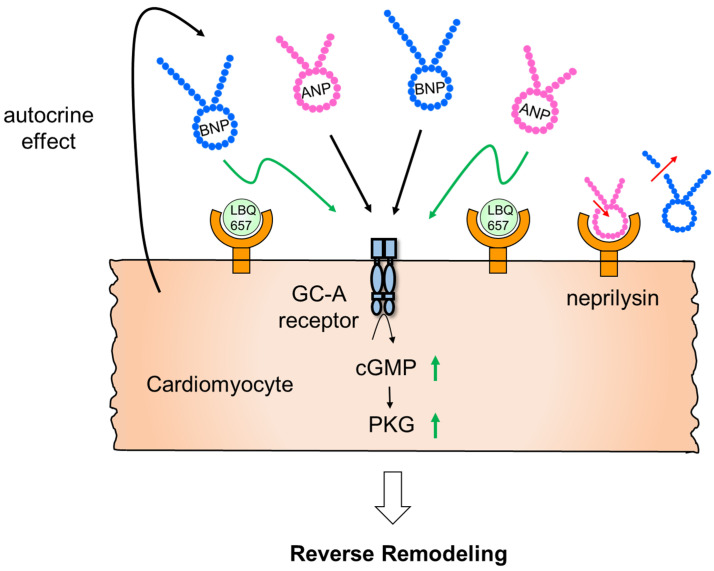
The effect of NEP on cardiomyocytes and the potential role of ARNI. Orally administered ARNI may directly affect the heart and lead to reverse remodeling by increasing cGMP signaling, even at high concentrations of cardiac natriuretic peptides [61]. LBQ657, sacubitrilat (the active metabolite of sacubitril).

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
