# Peer review of "Roles of Natriuretic Peptides and the Significance of Neprilysin in Cardiovascular Diseases"

_biology, 2022, doi:10.3390/biology11071017_

Round 1

Reviewer 1 Report

General Considerations

This a concise and interesting review focused on physiological roles of natriuretic peptides revealed in mice carrying GC-A or GC-B deletion, the significance of NEP in the metabolism of natriuretic peptides, and the long-term effects of ARNI. Although the review is focused on the results of experimental rather than clinical studies, however the evidences reported in this article can be very useful even for clinicians. The Figures further improve the understanding of the text. I have to address to the Authors only a minor point in order to further improve the scientific message of this interesting review.

Minor Points

1.     References. Author should more accurately revise the 67 article references reported at the end of the article. The first and second manes of the Authors are reported in different manner in some references.

Author Response

Our answers to the comments of reviewer #1:

We would appreciate you very much for evaluating our manuscript and checking the detail.

Reviewer’s question 1:  References. Author should more accurately revise the 67 article references reported at the end of the article. The first and second manes of the Authors are reported in different manner in some references.

Answer: Thank you for your checking in detail. We had changed the name of authors of references in the same way.

Reviewer 2 Report

Nakagawa and Saito describe the potential physiological role of neprilysin and natriuretic peptides. The review deals with an interesting topic due to the great effect Entresto has shown in clinical settings despite its not conclusively shown biochemical mode of action.

- The review is put in a good logical order to first deal with natriuretic peptides and neprilysin in general and than focussing on the effect of inhibition in different diseases/situations.

- Figure 2 does not reveal significant information and should be deleted.

- Instead there could be some kind of a fact sheet about neprilysin with its molecular characteristics and biochemical effects

- Please check again for spelling and grammar mistakes, there are several singular/plural errors concerning nouns and the respective verbs.

- Figure 3 -> Please also add the mode of actions for "reverse remodelling". Which pathways are up-or downregulated?

- Idea for another figure: Mode of action of different heart failure agens (MRA, ACE/AT1/ARNI, BB...) to illustrate the similarities and differences between different drugs.

Author Response

Our answers to the comments of reviewer #2:

We deeply appreciate you for sparing your precious time and giving us kind and constructive comments on our paper. We feel that we could make our paper more completed thanks to your supportive advice. Followings are our answers to your comments.

Reviewer’s question 1:  Figure 2 does not reveal significant information and should be deleted.

Answer: You are right. Figure 2 is not significant information. We deleted figure 2 from the article.

Reviewer’s question 2: Instead there could be some kind of a fact sheet about neprilysin with its molecular characteristics and biochemical effects

Answer: Thank you for your advice. As you pointed, many peptides are involved in the action of neprilysin. This article is focused on natriuretic peptides only. Therefore, we put the sentence below regarding characteristics of neprilysin against 3 natriuretic peptides.

(please see page 10, line 10)

These different biochemical effects of neprilysin on 3 natriuretic peptides may be one of the factors affecting the different half-lives of them.

Reviewer’s question 2: Please check again for spelling and grammar mistakes, there are several singular/plural errors concerning nouns and the respective verbs.

Answer: Thank you very much for your kind advice. We could correct my several mistakes.

Reviewer’s question 3: Figure 3 -> Please also add the mode of actions for "reverse remodelling". Which pathways are up-or downregulated?

Answer: You are right. Linked to figure1, we added following sentences (below question 4).

Reviewer’s question 4: Idea for another figure: Mode of action of different heart failure agens (MRA, ACE/AT1/ARNI, BB...) to illustrate the similarities and differences between different drugs.

Answer: Thank you for your kind advice. We would like to focus on natriuretic peptides in this article. Therefore, we added the following sentences regarding MRA and ARB linked to natriuretic peptides.

(please see page 11, the last sentences)

These protective actions are possibly affected by inhibiting MR, AT1R and TRPC in heart tissue. However, further studies are needed whether augmentation of natriuretic peptides signaling is enough to suppress MR and AT1R activity similar to MR antagonist and ARB in the heart.

Reviewer 3 Report

This paper described the physiological roles of natriuretic peptides elucidated in mice with GC-A or GC-B deletion, the significance of NEP in natriuretic peptide metabolism, and the long-term effects of angiotensin receptor-neprilysin inhibitor on cardiovascular diseases, which made a significant contribution to this field. I think some minor issues should be addressed before it can be processed.

1. Is "Simple summary" section necessary for this paper?

2. Physiological roles of natriuretic peptides were just only discussed from the aspects of genetically modified mice. Clinical studies and their current application in clinical diagnosis or prognosis evaluation should be also discussed. 

3. This review mainly introduced roles of natriuretic peptides and the significance of neprilysin from the perspective of cardiovascular disease. Therefore, I suggest changing the title to Roles of Natriuretic Peptides and the Significance of Neprilysin in Cardiovascular Diseases, which seems to be more accurate.

Author Response

Our answers to the comments of reviewer #3:

We would like to appreciate you very much for kind and honest comments. We think that it took a lot of time for you to give us your supportive comments.

Reviewer’s question 1: Is "Simple summary" section necessary for this paper?

Answer: Thank you for your suggestion. Simple summary is needed according to MDPI journal policy.

Reviewer’s question 2: Physiological roles of natriuretic peptides were just only discussed from the aspects of genetically modified mice. Clinical studies and their current application in clinical diagnosis or prognosis evaluation should be also discussed. 

Answer: You are right. We added a sentence regarding a clinical study and current application for clinical diagnosis as a biomarker.

(please see page 4, line 6)

Since the secretion of ANP and BNP from the failing heart is increased, plasma levels of ANP and BNP are clinically used as a biomarker to diagnose heart failure.

Reviewer’s question 3: This review mainly introduced roles of natriuretic peptides and the significance of neprilysin from the perspective of cardiovascular disease. Therefore, I suggest changing the title to Roles of Natriuretic Peptides and the Significance of Neprilysin in Cardiovascular Diseases, which seems to be more accurate.

Answer: Thank you for your important advice. We changed the title as you suggested.

Round 2

Reviewer 2 Report

The authors have adequately addressed my concerns, so that to my mind the manuscript can be recommended for publication in its present form.

Author Response

We would like to appreciate for your enormous effort to evaluate our manuscript. We believe that our manuscript was improved thanks to your important suggestions. Thank you very much.

Reviewer 3 Report

The authors have addressed all the issues I commented.

Author Response

(The authors gave the same response as above.)
